# Exposure to environmental pollutants selects for xenobiotic-degrading functions in the human gut microbiome

Francesca De Filippis [1,2,3], Vincenzo Valentino [1], Giuseppina Sequino[1], Giorgia Borriello[3], Marita Georgia Riccardi[3], Biancamaria Pierri[4], Pellegrino Cerino[4], Antonio Pizzolante[4], Edoardo Pasolli[1,2], Mauro Esposito[4], Antonio Limone[3] & Danilo Ercolini [1,2] ✉

Environmental pollutants from different chemical families may reach the gut microbiome, where they can be metabolized and transformed. However, how our gut symbionts respond to the exposure to environmental pollution is still underexplored. In this observational, cohort study, we aim to investigate the influence of environmental pollution on the gut microbiome composition and potential activity by shotgun metagenomics. We select as a case study a population living in a highly polluted area in Campania region (Southern Italy), proposed as an ideal field for exposomic studies and we compare the fecal microbiome of 359 subjects living in areas with high, medium and low environmental pollution. We highlight changes in gut microbiome composition and functionality that were driven by pollution exposure. Subjects from highly polluted areas show higher blood concentrations of dioxin and heavy metals, as well as an increase in microbial genes related to degradation and/or resistance to these molecules. Here we demonstrate the dramatic effect that environmental xenobiotics have on gut microbial communities, shaping their composition and boosting the selection of strains with degrading capacity. The gut microbiome can be considered as a pivotal player in the environment-health interaction that may contribute to detoxifying toxic compounds and should be taken into account when developing risk assessment models. The study was registered at ClinicalTrials.gov with the identifier NCT05976126.

The gut microbiome is increasingly recognized as important in influencing human health[1,2], and it has become clear that the microbiome and host mutually interact in an intimate relationship. The array of all the genes in the microbiome along with the host genes has been defined as the "hologenome"[3]. While the host genome is highly conserved and genetically adapts slowly to changes in the environment, the microbiome genome can rapidly change in response to the environment, and it has been indicated as a possible additional way of promoting evolution[4,5]. The fascinating hypothesis of the coevolution of the human microbiome with its host was proposed and discussed by several authors[6,7]. Indeed, a parallel evolutionary history was hypothesized for several bacterial species, suggesting that different strains coevolved with the host mirroring human migration patterns[7]. Although the rules driving this coevolutionary mechanism are unknown, host genetics, diet and exposure to other environmental factors can be considered the main influencers. Humans are exposed

[1]Department of Agricultural Sciences, University of Naples Federico II, Via Università, 100, Portici, Italy. [2]Task Force on Microbiome Studies, University of Naples Federico II, Corso Umberto I, 40, Napoli, Italy. [3]Istituto Zooprofilattico Sperimentale del Mezzogiorno, Via Salute, 2, Portici, Italy. [4]National Reference Centre for the Analysis and Study of the Correlation between Environment, Animal and Human, Via Salute, 2, Portici, Italy. ✉e-mail: ercolini@unina.it

daily to a wide range of xenobiotics that may include different classes of molecules (e.g., food additives, drugs, pesticides, process contaminants, and environmental pollutants) that can enter the human body through multiple routes, including skin contact, inhalation and ingestion[8]. These substances can reach the gut microbiome directly or can be previously metabolized in the liver[8]. For example, many different bacterial species have the potential to metabolize and chemically modify drugs, influencing their effects or the host metabolic response to the treatment[9,10]. In urbanized areas, several environmental pollutants from different chemical families may reach the gut microbiome, such as bisphenols, phthalates, polycyclic aromatic hydrocarbons, persistent organic pollutants (POPs; e.g., dioxins) and heavy metals. Several studies in animal models have suggested that exposure to these chemicals may lead to changes in the composition of the gut microbiome[11]. In addition, several microorganisms harbor an enzymatic repertoire conferring the ability to degrade and transform chemical pollutants, possibly detoxifying them or increasing their toxicity[12]. It is widely recognized that the presence of xenobiotics in the soil may drive genomic adaptation in the microbiome, boosting random mutations or gene acquisition events that can increase the resistance to toxic chemicals or the ability for their degradation[13]. Indeed, a catalog of microbial biocatalytic reactions of different environmental pollutants was developed, including >1500 reactions towards approximately 1400 compounds[14]. While these activities have gained much attention in soil microbes, fostering the application of pollutant-degrading strains for soil bioremediation[15], less is known about the gut microbiome. However, gut microbes have an extensive metabolic toolkit to degrade and/or transform environmental chemicals. Enzymes such as azoreductases, nitroreductases, β-glucuronidases, sulfatases and β-lyases, which are involved in the metabolism of >30 environmental pollutants, have been identified in the gut microbiome[12]. Therefore, gut microbial metabolism of pollutants may modulate their toxicity to the host.

Chemicals used in industrial and agricultural practices are leading to widespread contamination, and their effects on human, animal and environmental health are a major concern[16]. Epidemiological studies have highlighted that exposure to these pollutants may contribute to the development of several chronic diseases, including metabolic syndrome, type 2 diabetes[17], cardiovascular diseases[18] and cancer[19]. In addition to the environmental pressure due to anthropic activities typical of all developed countries, some areas in the Campania region (southern Italy) have attracted media coverage in the past 15 years, and are considered a dramatic example of an extremely polluted area[20]. The environmental pollution issue has become a significant concern in the Campania Region as a result of the "waste management crisis" that mainly affects the northern part of the region, encompassing 91 municipalities[21]. As a consequence, in the last few decades, a large rural area between the provinces of Naples and Caserta, primarily used for agriculture and livestock breeding, was considered to be at high risk of contamination due to the illegal disposal of urban and industrial waste[22]. In these landfills, a broad range of hazardous wastes from different parts of Italy has been found. In addition, the wastes have often been open-air burned, leading to this area being named the "Land of Fires"[23]. Senior and Mazza[24] firstly highlighted the high incidence of cancer deaths in a specific area of the Campania region (compared to regional and national rates), which was identified by the authors as the "triangle of death". Afterwards, several studies reported a link between illegal waste disposal and an increased risk of cancer for the population[22,25], potentially associated with human exposure to carcinogenic substances such as dioxins, dioxin-like compounds, or polycyclic aromatic hydrocarbons, which can be released into air, soil, and water bodies through the illegal dumping and burning of waste[25]. The concerns about population health and the safety of agricultural products cultivated in this area prompted local authorities to promote monitoring programs to assess the level of contaminants in soil,

groundwater, vegetables, animals and animal products, as well as a widespread screening of the population living in the "Land of Fires"[26,27]. As part of this translational study, a human biomonitoring study was carried out aimed at establishing how environmental pollution, food and lifestyles can affect residents' health. The scientific protocol (SPES – Exposure Study on Susceptible People; http://spes.campaniatrasparente.it) encompassed the evaluation of a set of biomarkers to quantify exposure and detect individual molecular responses to potential damage[28].

Indeed, this area was considered the perfect field for "exposomics" research[23]. Exploring the gut microbiome of subjects living in the "Land of Fires" can be an unprecedented opportunity to unravel the effects of exposure to multiple pollutants on gut microbes, to assess the toxicological relevance of the bacteria–xenobiotic interplay for the host and to determine how xenobiotic-microbiome interactions play a role in driving the adaptation process of the microbiome to the environmental pressure. In this study, we evaluated gut microbiome composition and functional potential in subjects constantly exposed to different levels of environmental pollution and highlighted the selection of specific microbiome signatures driven by pollution.

## Results
### Altered gut microbiome composition in subjects exposed to high environmental pollution

We analyzed the gut microbiome of 359 subjects living in different areas of the Campania region (southern Italy), divided into HIGH ($n = 82$), MEDIUM ($n = 161$) and LOW ($n = 116$) environmental pollution groups[29]. The groups did not differ in sex, age, body mass index, smoking habits, antibiotics use in the past 6 months (Supplementary Data 1), or dietary habits (Supplementary Fig. 1). Microbial diversity did not differ between the groups ($p > 0.05$), while the overall taxonomic composition of the gut microbiome was significantly different (PERMANOVA, $p < 0.05$), with differentiation of subjects in the HIGH and LOW groups (Fig. 1a). Consistently, Jaccard's (Fig. 1b) and Aitchinson's (Supplementary Fig. 2) distances within the HIGH and LOW groups was significantly lower than the distance between the groups, while the MEDIUM group showed higher variability (Fig. 1b and Supplementary Fig. 2). In addition, 115 taxa differed in relative abundance between HIGH and LOW, while 98 and 67 showed different relative abundances between MEDIUM and HIGH/LOW, respectively (Supplementary Data 2). The HIGH group had higher levels of Actinomycetota (previously classified as Actinobacteria[30]) than both the LOW and MEDIUM groups, while Pseudomonadota (previously classified as Proteobacteria[30]) dominated in the MEDIUM group ($p < 0.05$; Fig. 1c, d). In particular, several species of Actinomycetota identified as *Eggerthellaceae*, *Actinomyces* spp., *Bifidobacterium* spp. and *Corynebacterium* sp. were enriched in the HIGH group, as well as several unidentified Clostridia and *Ruminococcaceae* species ($p < 0.05$; Supplementary Data 2). In contrast, some unidentified *Bacteroides* species and several metagenomic OTUs (mOTUs) identified as *Faecalibacterium* spp. were depleted in the HIGH group compared with the LOW group, whereas several mOTUs taxonomically identified as *Pseudomonas* spp. (phylum Pseudomonadota) were significantly enriched in the MEDIUM group (Supplementary Data 2). We used a machine-learning-based classification approach to evaluate if the gut microbiome composition at species-level could discriminate subjects of the three groups. We observed a moderate (area under the curve AUC = 0.71, 95% Confidence Interval, CI: 0.66–0.76) but significant ($p < 0.01$ by computing the statistical test against the null hypothesis of equal AUC for classification of true and shuffled labels) discrimination between LOW and HIGH groups (Supplementary Fig. 3A). Moreover, we found a high discrimination (AUC = 0.77 and 0.83, respectively) when comparing MEDIUM vs LOW or HIGH, supporting the finding that specific microbial signatures are associated with the level of exposure to environmental pollutants. Consistently, we identified

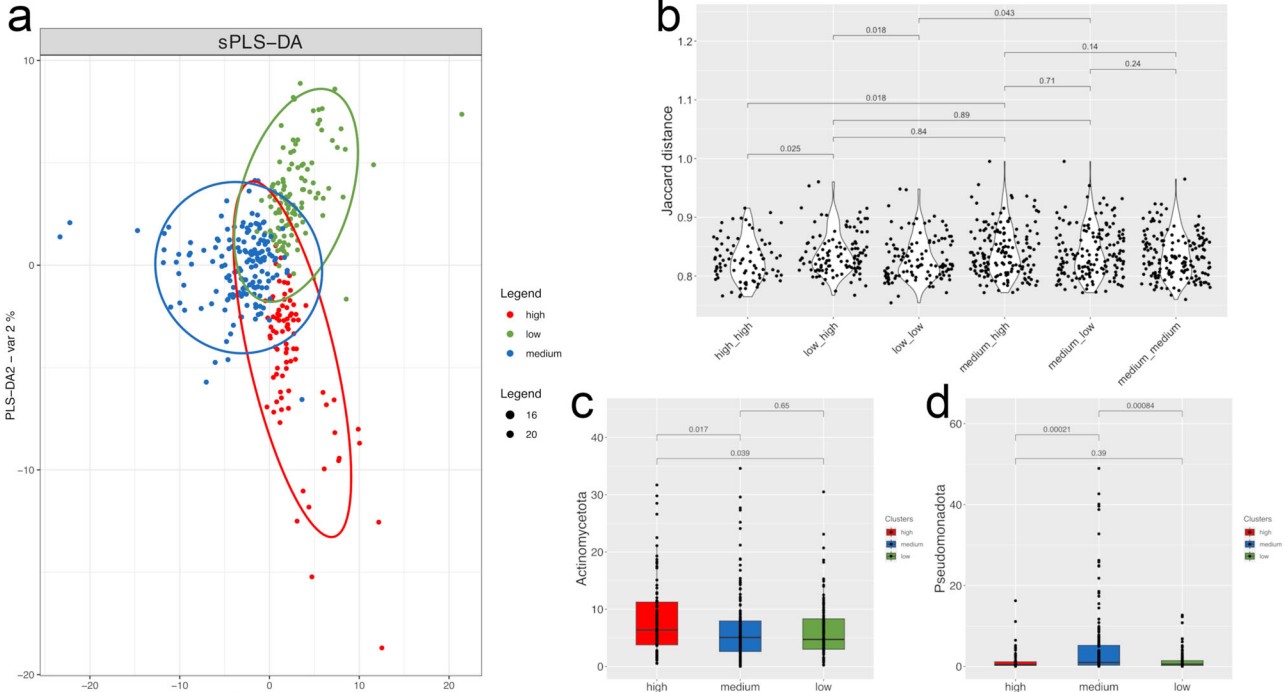

**Fig. 1 | Exposure to environmental pollution drives changes in gut microbiome composition.** Sparse Partial Least-Squares Discriminant Analysis (sPLS-DA) model based on microbiome taxonomic composition (**a**). Violin plots showing Jaccard's distance based on gut microbiome composition between groups (**b**); Box plots showing the relative abundance (%) of Actinomycetota (**c**) and Pseudomonadota (**d**) in subjects from areas at HIGH, MEDIUM and LOW environmental pressure, as defined by MIEP index. Boxes represent the interquartile range (IQR) between the first and third quartiles, and the line inside represents the median (2nd quartile). Whiskers denote the lowest and the highest values within 1.5 ×IQR from the first and third quartiles, respectively. The significance was tested by applying pairwise, two-sided Wilcoxon test. Data are obtained from $n = 82$, 161 and 116 biologically independent samples from HIGH, MEDIUM and LOW environmental pollution groups, respectively.

again several *Faecalibacterium* mOTUs (e.g., *Faecalibacterium* species *incertae sedis* mOTU_12325, 12403, 12331 and *F. prausnitzii* mOTU_06112, 06110), *Bacteroides coprocola* (mOTU_11279) and *Bacteroides uniformis* (mOTU_00855) among the most discriminant features between HIGH and LOW groups (Supplementary Fig. 3B).

Finally, we used *mmvec* to evaluate correlations of taxa relative abundance with dioxins and heavy metals blood concentration. Interestingly, we found that the plasmatic levels of polychlorinated dibenzo-para-dioxins/furans and dioxin-like polychlorinated biphenyls were negatively associated with two mOTUs identified as *Lachnospiraceae* and *Ruminococcus gnavus* (Supplementary Fig. 4A). On the contrary, two mOTUs taxonomically identified as *Pseudomonas* spp. (phylum: Pseudomonadota) and significantly more abundant in the MEDIUM group (Supplementary Data 2) were positively linked with the plasmatic levels of mercury (Hg) and antimony (Sb; Supplementary Fig. 4B).

### Exposure to environmental pollutants selects for xenobiotic-degrading ability in the gut microbiome

Increased gene richness was observed in the HIGH group, indicating that exposure to high environmental pressure may lead to the acquisition of novel potential functions (Supplementary Fig. 5). Indeed, several genes related to xenobiotic degradation or resistance were enriched in the HIGH and MEDIUM groups compared with the LOW group, including genes coding for dehalogenases and chlorobenzene dioxygenases, which are involved in the degradation of different dioxin classes (Fig. 2a). Accordingly, serum concentrations of polychlorinated dibenzo-para-dioxins/furans, polychlorinated dibenzofurans, polychlorinated-p-dioxins/furans and dioxin-like polychlorinated biphenyls were higher in the subjects of the HIGH and MEDIUM groups (Fig. 2b). Interestingly, the plasmatic concentration of these compounds was

positively correlated with the relative abundance of several genes related to dioxin degradation in the gut metagenomes (Supplementary Figs. 6 and 7). In addition to dioxin-related genes, the gut microbiomes of HIGH subjects were also enriched in genes related to the degradation of other types of organic environmental pollutants, such as 1-carboxy-3-chloro-3,4-dihydroxycyclo hexa-1,5-diene dehydrogenase, which is involved in the degradation of polyethylene terephthalate (PET), and salicylaldehyde dehydrogenase, which is related to the naphthalene and anthracene degradation pathways (Fig. 2a). In addition, benzoate dioxygenase, involved in the metabolism of the food additive benzoate, was also enriched in HIGH subjects, suggesting that exposure to environmental organic pollutants may select for a wide range of activities against compounds with similar chemical structures (Supplementary Fig. 8). Finally, we screened metagenomes for the presence of genes related to adaptation to heavy metals using the BacMet database. Several genes related to heavy metal transport, efflux and resistance were identified in the gut metagenomes. Surprisingly, most of these genes were enriched in the MEDIUM group (Fig. 3a), which also showed higher blood concentrations of several metals than the LOW group and, in many cases, even than the HIGH group (Fig. 4). Indeed, this result was mainly driven by sub-clusters 19, 20, and 21, which showed higher metal levels than other subclusters in the MEDIUM group and than the HIGH group (Supplementary Fig. 9). The MEDIUM group also showed a higher abundance of several genes related to multidrug efflux and resistance, possibly involved in antibiotic and biocide compound resistance (Fig. 3b).

### Pollutant-related genes are more abundant in disease compared to healthy cases

We further investigated the diffusion of genes related to POPs and metal degradation/resistance by looking for these genes in publicly

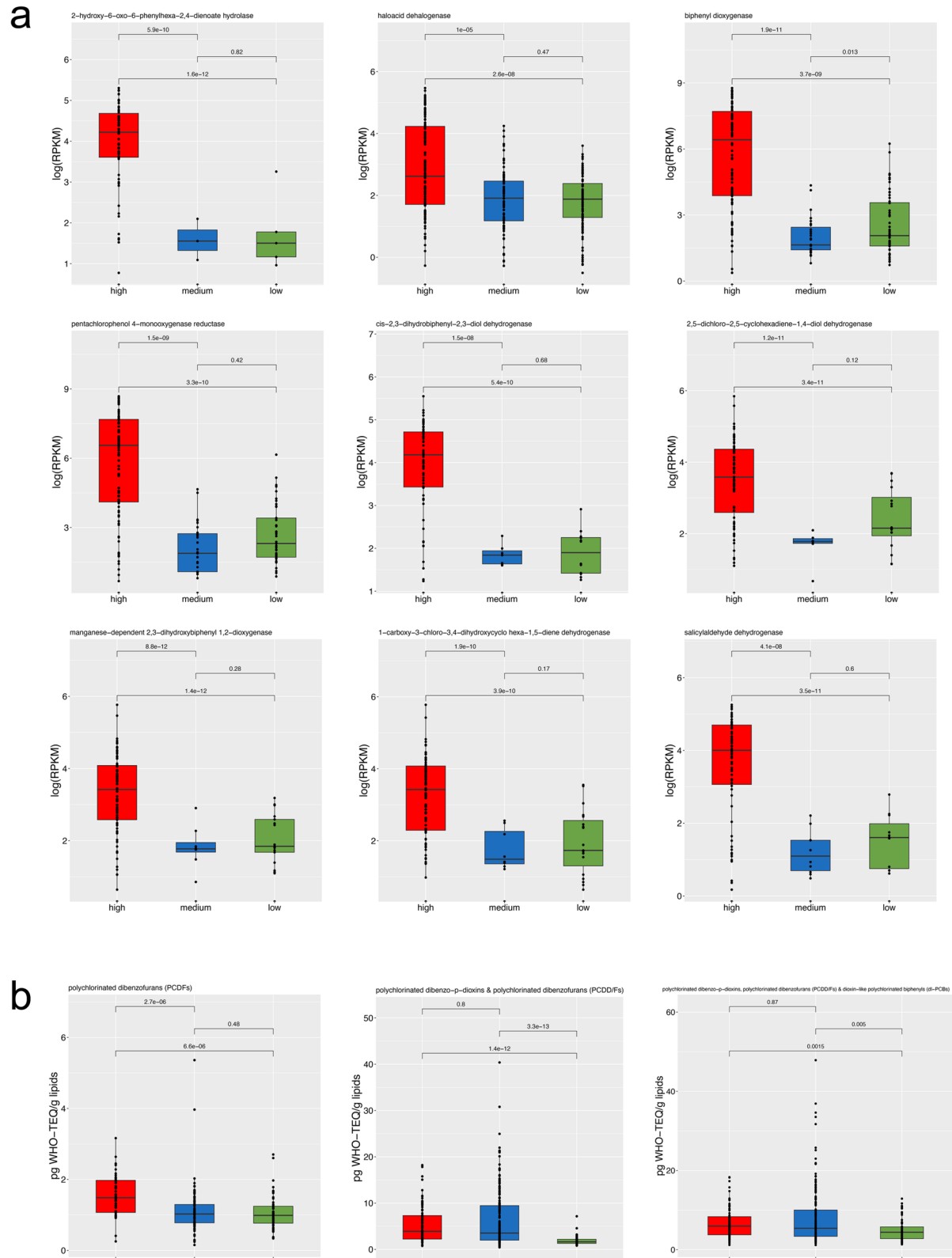

**Fig. 2 | Dioxin exposure drives the enrichment of a potentially degrading pattern in the gut microbiome.** Box plots showing the abundance of genes coding for dioxin degradation enzymes (log Reads Per Kilobase per Million, RPKM; **a**) and the plasmatic concentration of different dioxin classes (**b**) in subjects from areas at HIGH, MEDIUM and LOW environmental pressure, as defined by MIEP index. Boxes represent the interquartile range (IQR) between the first and third quartiles, and the line inside represents the median (2nd quartile). Whiskers denote the lowest and the highest values within 1.5 ×IQR from the first and third quartiles, respectively. The significance was tested by applying pairwise, two-sided Wilcoxon test with *p*-value correction using the False Discovery Rate approach. Data are obtained from *n* = 82, 161 and 116 biologically independent samples from HIGH, MEDIUM and LOW environmental pollution groups, respectively.

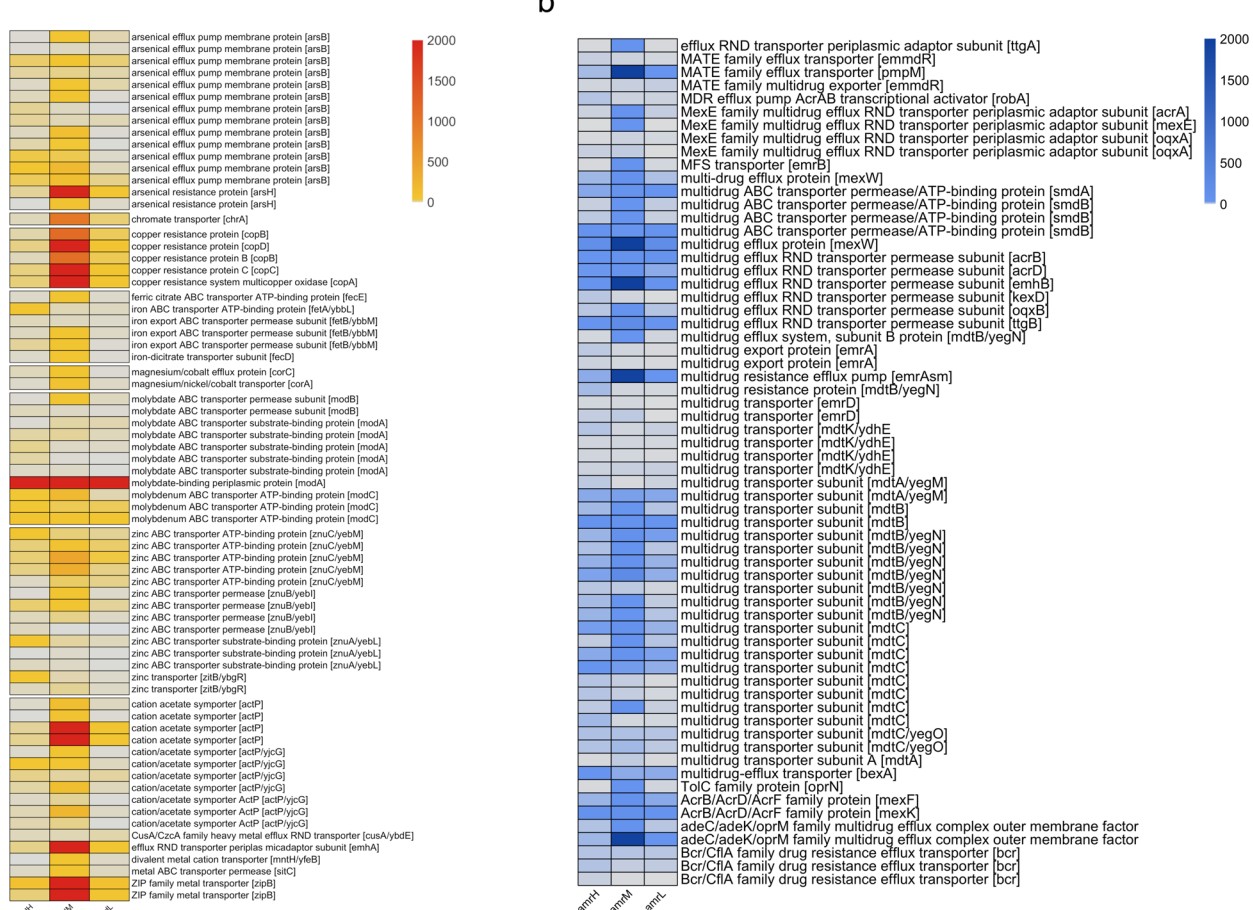

**Fig. 3 | Heavy metals exposure selects for antibiotic resistance genes.** Heatplots showing the abundance of genes coding for heavy metals resistance and transport enzymes (**a**) and antimicrobial resistance genes (**b**) in subjects from areas at HIGH, MEDIUM and LOW environmental pressure, as defined by MIEP index. Values are represented as mean log Reads Per Kilobase per Million (RPKM) from $n = 82$, 161 and 116 biologically independent samples from HIGH, MEDIUM and LOW environmental pollution groups, respectively.

available gut metagenomes. We considered a total of 3769 subjects from 24 datasets, including case–control studies composed of healthy adult controls and subjects with different diseases (liver cirrhosis, $n = 282$ subjects; colorectal cancer, CRC, $n = 1395$; hypertension, $n = 235$; inflammatory bowel disease, IBD, $n = 851$; metabolic syndrome, $n = 15$; type-1 and type-2 diabetes, $n = 173$ and 818). Genes predicted from these gut metagenomes were mapped against the BacMet and mibPOP databases as described before. Interestingly, the number of hits was significantly higher in different diseases (e.g., cirrhosis, IBD, CRC; Supplementary Fig. 10A). In addition, the prevalence of haloacid dehalogenase (AVT80512, involved in the degradation of halogenated pollutants, such as PCBs), salicylaldehyde dehydrogenase (AAP44246, which participates in naphthalene and anthracene degradation) and several genes involved in metal transport and resistance was higher in these patients than in healthy controls (Supplementary Fig. 10B).

**Environmental pollution selects for xenobiotic-degrading strains and promotes antibiotic resistance gene acquisition**
We reconstructed a total of 12,569 high/medium quality Metagenome Assembled Genomes (MAGs) from the gut metagenomes, classified into 947 SGBs (Species-level Genome Bins; Supplementary Data 3). Among them, 904 SGBs (including 12,523 MAGs, >99% of the MAGs retrieved) showed <5% distance from genomes reported in the database (kSGBs). Bacillota ($n = 9388$), Bacteroidota ($n = 1365$), Actinomycetota ($n = 1192$) and Pseudomonadota ($n = 326$) were the most

represented phyla (Supplementary Fig. 11). Interestingly, genes related to dioxin degradation were identified in 4 Actinomycetota SGBs, all belonging to the *Eggerthellaceae* family, identified as *Asaccharobacter* sp. (SGB318), *Eggerthella lenta* (SGB319), *Gordonibacter urolithinfaciens* (SGB320), and *G. pamelaeae* (SGB321) (Fig. 5a). In particular, all MAGs belonging to SGB320 and SGB321 retrieved from HIGH and MEDIUM subjects harbored L-2-haloacid dehalogenase, while chlorobenzene dioxygenase was found in most of the *Escherichia coli* MAGs reconstructed, with a higher prevalence (82.6%) in those retrieved from the HIGH group (Fig. 5a). Several SGBs also showed genes coding for metal efflux pumps, metal transport and resistance (Fig. 5b). Consistently, MAGs reconstructed from MEDIUM subjects showed a higher prevalence of these genes, as well as of genes involved in resistance to antimicrobial compounds (Fig. 5c). Indeed, 242 MAGs harbored both antimicrobial and metal resistance genes, with a higher prevalence in those reconstructed from MEDIUM subjects, and interestingly, most of these MAGs belonged to Pseudomonadota (Fig. 6).

## Discussion
Humans are constantly exposed to the so-called "exposome", defined as the complex entirety of environmental factors we are exposed to ref. 31. Among them, pollution can be considered one of the most important. In this context, our microbial symbionts play a dual role, since they are exposed to the same factors as the host, but, at the same time, they may be considered part of the exposome, metabolizing environmental pollutants and producing other molecules[32].

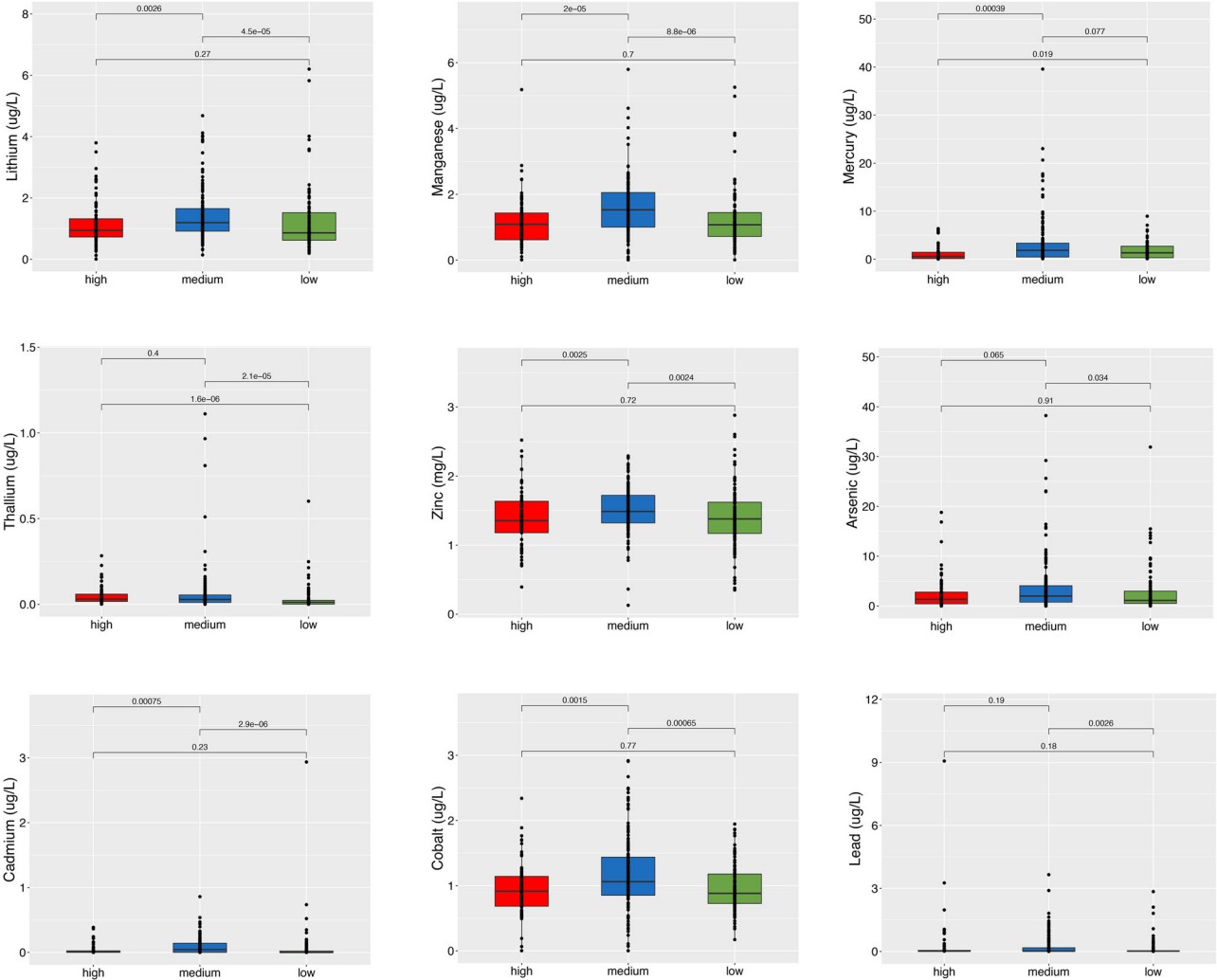

**Fig. 4 | Heavy metal concentration is higher in subjects from the MEDIUM environmental pressure area.** Box plots showing the plasmatic concentration of different heavy metals in subjects from areas at HIGH, MEDIUM and LOW environmental pressure, as defined by MIEP index. Boxes represent the interquartile range (IQR) between the first and third quartiles, and the line inside represents the median (2nd quartile). Whiskers denote the lowest and the highest values within 1.5 ×IQR from the first and third quartiles, respectively. The significance was tested by applying pairwise, two-sided Wilcoxon test. Data are obtained from $n = 82$, 161 and 116 biologically independent samples from HIGH, MEDIUM and LOW environmental pollution groups, respectively.

Unraveling the complex interactions between the host, human microbiome and exposome is fundamental to understanding the impact of environmental factors on human health and to deepening the study of host-microbiome interplay.

In this study, we explored the influence of exposure to environmental pollution on the gut microbiome composition and potential activities. We selected as a case study a population living in a highly polluted area in the Campania region (southern Italy), proposed as the perfect area for exposomic studies[23]. Considering a previous classification of Campania municipalities in areas at high, medium and low environmental pressure[29], we compared the gut microbiome of subjects living in the three areas, providing striking evidence on the ability of the gut microbiome to taxonomically and functionally adapt to environmental pollution. Subjects living in areas characterized by greater environmental pressure showed higher plasmatic levels of dioxins and PCBs, which were correlated with changes in their gut microbiome. Prolonged exposure to pollution promoted the selection in the gut microbiome of several taxa belonging to Actinomycetota and Pseudomonadota, phyla often reported as able to degrade organic pollutants in soils and sediments[33,34] and proposed for soil bioremediation[35,36]. Moreover, the gut microbiome of subjects living in highly contaminated areas was also enriched in genes related to the degradation of POPs, as well as other organic complex molecules. This finding suggests that high exposure to these compounds may select for a gut microbiome that is equipped for their degradation. Indeed, the gut microbiome has been suggested to metabolize several xenobiotics, including food additives, drugs, pesticides and environmental chemicals[9,12,37].

Unexpectedly, we found higher concentrations of heavy metals in the blood of subjects living in an area previously classified as MEDIUM environmental impact compared with subjects from the HIGH impact area, particularly in subclusters 19, 20, and 21 (as defined based on MIEP[28,29]). This apparent discrepancy may be explained by considering the factors evaluated in the calculation of the MIEP, which did not include the influence of climate and geomorphological features. Indeed, the three subclusters 19, 20 and 21 in the MEDIUM area are characterized by a particular morphology (flat area enclosed by mountains) that limits air circulation and may explain a particular accumulation of toxic compounds in these subclusters. Accordingly, we also identified a higher abundance of genes related to metal transport and efflux in the gut metagenome of these subjects. Interestingly, we observed higher prevalence of genes related to antibiotic and antimicrobial resistance in MAGs from MEDIUM group, compared with both HIGH and LOW groups, although the consumption of

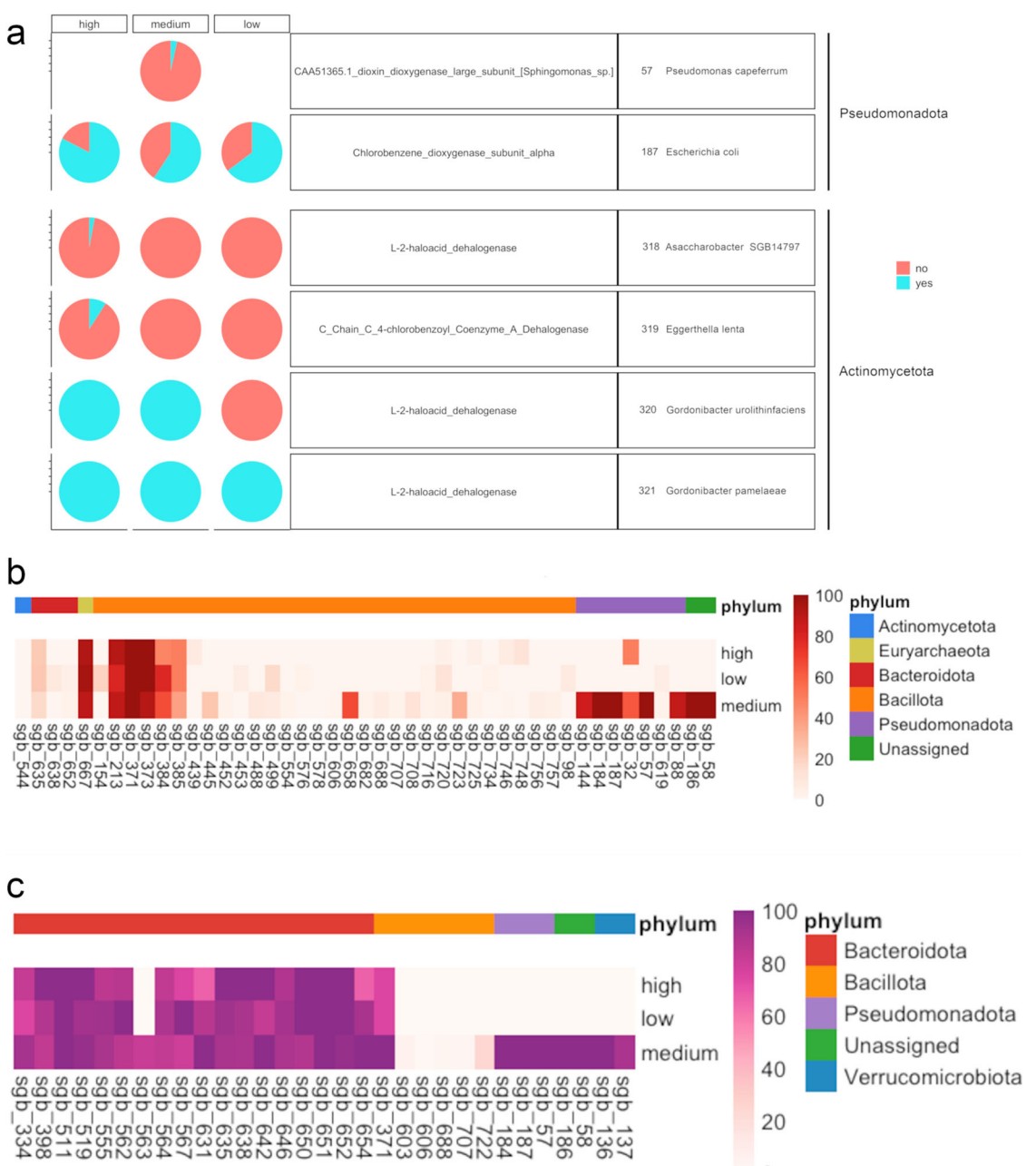

**Fig. 5 | Resistance genes are widespread in microbial genomes reconstructed from metagenomics reads.** Pie chart showing the proportion (%) of MAGs from each SGB where genes related to dioxin degradation pathways were identified. Yes, the gene was found; no, the gene was not found (**a**). Heatmap showing, for each SGB, the proportion (%) of MAGs where at least one gene related to heavy metal (**b**) and antimicrobial (**c**) resistance was identified. SGBs are color-coded according to the phylum-level taxonomic classification. Only SGBs with ≥10 MAGs are shown.

antibiotics in the three groups did not differ and only few subjects per group declared the use of antibiotics in the past 6 months. Indeed, it was previously reported that heavy metals and antibiotic resistances are co-selected in soils and sediments[38–40], either because the related genes are located near each other on the genomic elements or because the same gene is responsible for both resistances[41], making the impact of metal pollution even more alarming for public health.

Our results confirmed the dramatic effect that environmental xenobiotics have on microbial communities, shaping their composition and boosting the selection of strains with degrading capacity. Indeed, the effect of petroleum hydrocarbon, heavy metals and microplastic contamination on the soil and marine microbiome was reported[42,43]. Zrimec and colleagues[44] showed that the plastic pollution patterns in global soils and oceans correlate with the relative abundance and diversity of plastic-degrading microbial enzymes, demonstrating the plasticity of the gut microbiome, which can quickly adapt in response to external perturbations[45,46].

Conversely, the role of xenobiotics in the gut microbiome is less studied. It was recently highlighted that the gut microbiome can influence the metabolism of different nonantibiotic drugs, metabolizing and transforming them, thus affecting bioactivity and bioavailability[47]. However, the effects of xenobiotics from environmental pollution are still underexplored, as well as how this interplay can influence human health. We found that genes related to xenobiotic resistance/degradation were more abundant in disease cases, exploring a population of approximately 4000 subjects involved in case/

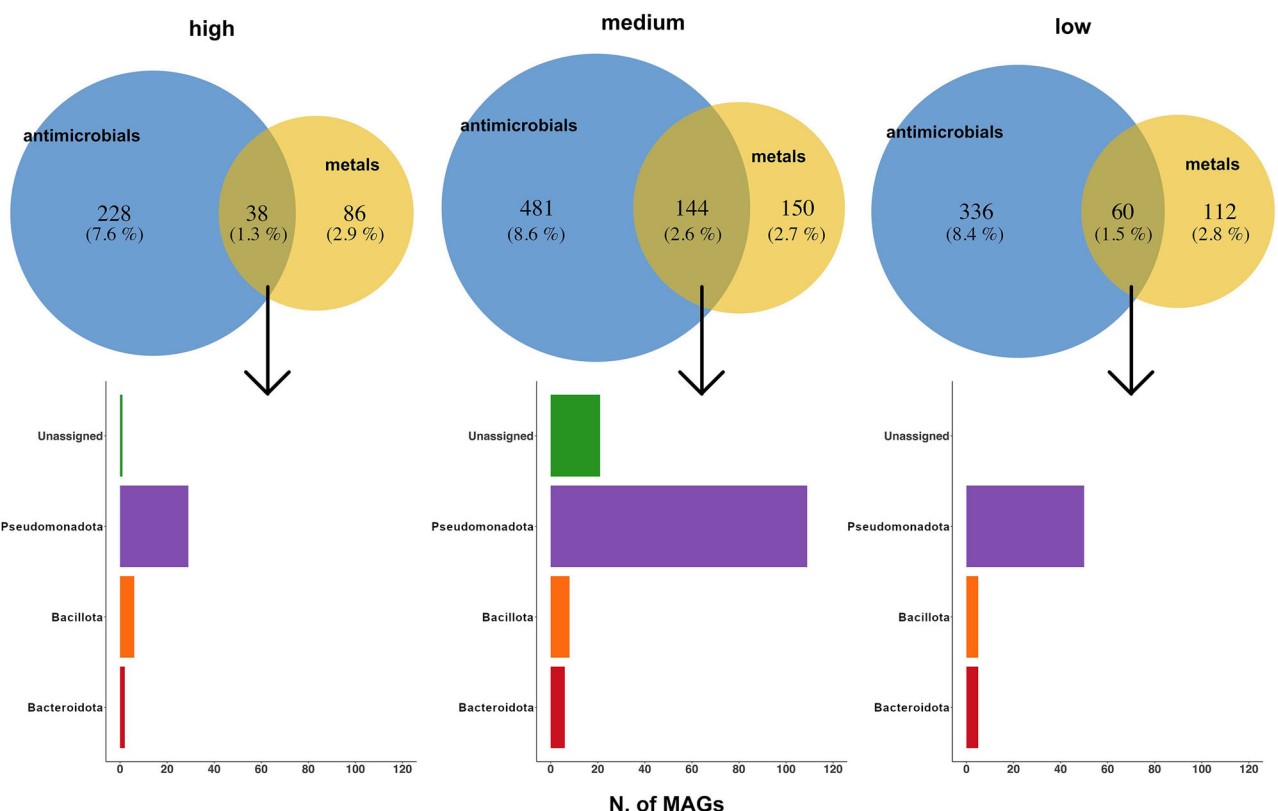

**Fig. 6 | Heavy metal and antimicrobial resistance genes are co-selected in the gut microbiome.** Venn diagrams showing the number of MAGs reconstructed from gut metagenomes of subjects from HIGH, MEDIUM and LOW environmental pressure groups harboring at least one antimicrobial and/or one heavy metal resistance gene. For the MAGs sharing both, a bar chart reporting the phylum-level ranking is shown.

control gut microbiome studies, which points to such genes as markers of disease. Indeed, gut microbiome adaptation to pollutants and its consequences are unknown, and the interplay between the occurrence of marker genes/species, exposure to pollutants and disease deserves in-depth investigation. Moreover, the interactions among multiple chemicals should also be evaluated. There is a growing interest in the involvement of pollutant exposure in the development of chronic diseases such as obesity, diabetes, and cancer and in the malfunction of the immune and reproductive systems. Our results demonstrated that the gut microbiome can be considered a pivotal player in these interactions, which may contribute to detoxifying toxic compounds and should be taken into account when developing risk assessment models. Upon further ad hoc designed investigations, the gut microbiome of subjects particularly exposed to environmental pollution may represent a source of xenobiotic-degrading bacteria for the development of novel probiotics that can potentially counteract the negative health effects of environmental pollution.

## Methods

### Ethics approval and consent to participate

The study was conducted in accordance with the Helsinki Declaration (Fortaleza revision, 2013), the Good Clinical Practice Standards (CPMP/ICH/135/95), the Italian Decree-Law 196/2003 regarding personal data, and the European regulations on this subject. On recruitment, participants signed a document in which they consented to the confidential treatment of their personal information, in accordance with current legislation on privacy (EU 679/2016) and consented to the publication of anonymized data. The study protocol, the subject information sheet and the informed consent form were reviewed and approved by the Ethics Committee of the IRCCS "G. Pascale" through Commissioner Deliberation n. 590 of the 3rd of August 2016. The study was registered

in the Clinical Trials Protocol Registration System at ClinicalTrials.gov with the identifier NCT05976126.

### Study population and sample collection

This observational, cohort study involved a subset of the population of healthy adults from Campania region recruited in the human biomonitoring study SPES, promoted by the Istituto Zooprofilattico Sperimentale del Mezzogiorno (IZSM) of Portici (Italy) in collaboration with the National Tumor Institute IRCCS (Istituto di Ricovero e Cura a Carattere Scientifico) "G. Pascale" in Naples. All subjects visiting the center in the considered period (September 2016 to August 2017) and meeting the inclusion criteria were recruited. Considering different source of environmental pressure (e.g., land use and population density, presence of contaminated sites, quality of air, soil and water bodies, closeness to waste management plants and illegal waste spills and fires), Pizzolante and colleagues[29] developed an integrated model to compute the Municipality Index of Environmental Pressure (MIEP). The MIEP takes into account all factors involved in the pollution processes, including the sources of contaminants and their migration pathways. The MIEP enables the identification of 21 homogeneous subclusters, which include several municipalities based on their environmental pressure scores. This categorization has resulted in three macro-areas with increasing levels of impact pressure: LOW, MEDIUM and HIGH, as described by Pierri and colleagues[28]. From September 2016 to August 2017, a total of 4227 subjects were enrolled in the SPES trial, considering healthy subjects living in several regional areas with different environmental pressures. A subgroup of 359 subjects residing in the three impact areas was randomly selected from the entire SPES cohort for this study, including all subjects providing the fecal sample. The inclusion criteria for the whole SPES cohort were: healthy subjects (no severe, chronic or neoplastic conditions), 20–50 years

old, both sexes (self-declared), no important viral infectious (HIV, HCV, HBV), no use of drugs, no alcohol abuse, no obesity of II, III and IV classes, and be resident in one of the municipalities under study for at least 5 years. Participants first underwent medical examination, completed a questionnaire on lifestyle and eating habits (EPIC questionnaire – European Prospective Investigation into Cancer and Nutrition - validated by WHO – World Health Organization[48]), and filled in an anamnestic (professional, familiar, clinical and pathological) case report form (CRF). Metadata collected, including age, sex, Body Mass Index (BMI), smoking habits, antibiotic use in the past 6 month are reported in Supplementary Data S1.

Peripheral blood (about 100 mL in several aliquots) was collected from the volunteers, in the early morning, in blood collection tubes (SST II Advance Tubes, BD Vacutainer). The samples were left for approximately 50 min, then the serum was separated from whole blood by centrifugation at $2000 \times g$ for 10 min at 4 °C and then aliquots were stored at −80 °C until analysis. Fecal samples were self-collected following the SOP04 developed by the International Human Microbiome Standard Consortium (www.microbiome-standards.org). Fecal samples were transported refrigerated within 2 h from the collection to the laboratory, where they were aliquoted and stored at −80 °C until the analyses.

### Analysis of dioxins (PCDD/Fs) and polychlorinated biphenyls (PCBs)

The determination of PCDD/Fs and PCBs in human blood serum was carried out using a modified analytical method described by Brasseur et al.[49] The method allowed the determination of 17 PCDD/F and 12 DL-PCB congeners showing the highest toxicity for humans[49,50] and of 6 NDL-PCBs (PCBs 28, 52, 101, 138, 153 and 180). More details are reported in the Supplementary Methods.

### Trace elements analysis

The determination of metals and metalloids was carried out on the serum obtained from the volunteers by ICP-MS NexION 350X (Perki-nElmer, Waltham, USA). The analytical panel of trace elements includes arsenic (As), beryllium (Be), cadmium (Cd), cobalt (Co), chromium (Cr), copper (Cu), iron (Fe), mercury (Hg), lithium (Li), manganese (Mn), molybdenum (Mo), nickel (Ni), lead (Pb), antimony (Sb), selenium (Se), strontium (Sr), thallium (Tl), vanadium (V), and zinc (Zn). More details are reported in the Supplementary Methods.

### Gut metagenome sequencing and data analysis

DNA extraction from fecal samples was carried out following the SOP 07 developed by the International Human Microbiome Standard Consortium (www.microbiome-standards.org) at the Department of Agricultural Sciences of the University of Naples Federico II (Portici, Italy). DNA libraries were prepared according to Illumina protocol and shotgun metagenomes were sequenced on NovaSeq platform, leading to 2x150bp, paired-end reads. Details on metagenomics data analysis are reported in the Supplementary Methods and assembly results are reported in Supplementary Data 4. A list of the Metagenome Assembled Genomes (MAGs) reconstructed is reported in Supplementary Data 3.

### Statistical analysis

No statistical method was used to predetermine sample size. The Investigators were not blinded to allocation during experiments and outcome assessment. No data were excluded from the analyses. Differences in the overall gut microbiome taxonomic composition according to the area of origin of the subjects (HIGH, MEDIUM, LOW) were assessed by PERMANOVA (Permutational Multivariate Analysis of Variance, *adonis* function, *vegan* R package) computed on Jaccard's and Aitchinson's distance matrices ($p < 0.05$). Comparisons of blood metabolites, taxa or gene relative abundance between groups were carried out using pairwise Wilcoxon tests with multiple-hypothesis

testing corrections via the false discovery rate (FDR). Distribution of smoking or antibiotic use in the cohort was evaluated using pairwise Fisher's tests ($p < 0.05$). Sparse Partial Least Squares Discriminant Analysis (sPLS-DA) on taxa relative abundance tables was computed using the R package *mixOmics*. Interactions between microorganisms and dioxins/metals were explored through *mmvec*[51] with default settings. Machine-learning-based classification analysis was done using the MetAML[52] package and by considering Random Forests (RFs) as back-end classifier for all the experiments. Results were obtained through a ten-fold cross-validation and averaged on 5 independent runs.

### Reporting summary

Further information on research design is available in the Nature Portfolio Reporting Summary linked to this article.

## Data availability

The raw sequence reads generated in this study have been deposited in the Sequence Read Archive (SRA) of the NCBI under accession number PRJNA861716. NCBI RefSeq genomes used in this study can be downloaded from https://ftp.ncbi.nlm.nih.gov/refseq/release/bacteria; human MAGs previously reconstructed[75] and used in this study can be downloaded from http:// segatalab.cibio.unitn.it/data/Pasolli_et_al.html.

## Code availability

All software used for analyses are publicly available for download.

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

## Acknowledgements

We would like to thank all the volunteers in our cohort and Federica d'Ascoli, Antonio di Stasio, Palmiero Volzone and Annachiara Coppola for their skillful assistance in samples collection and handling. This work was supported by the Italian Ministry of Health, with a grant to the projects *Linking environmental pollution and gut microbiota in individuals living in contaminated settlements* (Ricerca Finalizzata, GR-2016-02362975). The study was also supported by Campania Region (PAC Funding - Action B4).

## Author contributions

F.D.F. and D.E. conceived the study; P.C., A.P., B.P., M.E., A.L. designed the subject selection and recruitment; P.C., A.P., B.P., M.E. performed the sample collection and chemical analyses; G.S, V.V., M.G.R. and G.B. performed the metagenomics experiments; F.D.F., G.S., V.V., E.P., analyzed the data; F.D.F., D.E., P.C., M.G.R. acquired funding; F.D.F. wrote the manuscript. All authors read and edited the manuscript.

## Competing interests

The authors declare no competing interests.
