## [Peer Review File · Nature Communications]

REVIEWER COMMENTS

Reviewer #1 (Remarks to the Author):

Reviewers Comments

This study investigates the impact of environmental pollution on the gut microbiome in a population from a highly polluted area in Campania, Italy. It presents novel findings showing that exposure to pollutants like dioxins and heavy metals alters gut microbiome composition and increases the abundance of microbial genes related to pollutant degradation and resistance. This adaptation suggests the gut microbiome's potential role in detoxifying harmful compounds, highlighting its significance in environmental health interactions and risk assessment models. The research underscores the urgent need for further exploration of the microbiome's capacity to counteract pollution's adverse health effects, potentially leading to the development of probiotic treatments.

Major Comments:

The differences in the microbiome betadiversity between the 3 populations is quite striking, I worry about other drivers beyond pollution – while I agree that the communities appear to be relative homogeneous. I would like to see a more robust analysis of the microbial differences using more appropriate statistical tests to account for structural differences between the cohorts (see below).

It would be good to determine if the microbiome is predictive of burden – and what those features are, i.e. Random Forest. Also, I would like to see some attempt to derive predictive assessment with AuROC analysis to determine if the microbial features are predictive of the pollutant burden. It would also be nice to see some attempt as using tools like mmvec (<https://www.nature.com/articles/s41592-019-0616-3>) to really determine if the pollutant levels in blood are associated with specific MAGs or pathways.

Minor Comments:

Ln 87-91 – this paragraph feels like it belongs earlier in the justification for this introduction.

Ln 141 – this is the first report of ‘mOTUs’ so please define.

Ln 130-131 – “with clear clustering of the HIGH and LOW 130 groups (Figure 1A)” – maybe I am missing something, but I do not see any ‘clear clustering in this ordination, if anything the communities looks fundamentally different to each other? Can you run this as Aitchinson’s distance and RobustPCA to see if these differences hold up for Fig. 1b?

Ln 133-142 – it is not clear in Fig 1 C and D what the ‘abundance metric’ is for the Y axis. And you really should not use Wilcoxon for compositional data. If you did a CLR-transform you could maybe argue that it works, but it’s better to look at log-ratios with something like ANCOM BC or SongBird? Also why do this at the Phylum level, why not run a proper OGU analysis (<https://journals.asm.org/doi/full/10.1128/msystems.00167-22>) and then use a log-ratio analysis to identify those taxa that are differentially proportionally different. Even better run Random Forest and find the taxa that best describe the differences between these communities. I would also suggest the same for the functional gene analysis in Fig 2. Also please note that all your data are proportional and so try and refrain from using the word ‘abundance’, unless you put relative in front of it.

Reviewer #2 (Remarks to the Author):

In general, De Filippis and colleagues explored a fantastic cross-sectional population cohort in the Campania region of Italy covering different exposures to environmental pollution grouped in HIGH, MeEDIUM and LOW. Fecal samples and blood samples were retrieved from 359 healthy subjects randomly selected from the SPES trial (4227 subjects). Trace elements and dioxins were determined in blood. Fecal microbiome profiling showed compositional changes in response to the level of pollution and identified bacterial genes associated with dioxin degradation and antibiotic resistance. In summary, the study addresses the important question to which extent the environment affects microbiome changes, and the presented study cohort is clearly designed and suitable to address this question. While the data give valuable insights, the study remains descriptive and lacks validation.

Since only a sub-group of subjects in the SPES trial have been assigned to this study, additional validation seems possible in remaining smaller groups of subjects. Are compositional and functional changes reproducible and representative to the pollution categories. Would it be possible to validate response to selected chemicals in selected bacteria identified in the analysis?

In Figure 1, the authors use PLS-DA analysis to show differences between high, medium and low exposed subjects. However, PLS-DA is suitable to identify features that best describe differences between groups rather than differences in microbiota composition per se. I wonder how the

supervised and un-supervised PCoA analysis describes differences between these groups. Are the selected targets for further analysis justified by statistical selection or choices of the authors? The authors give no feature selection including feature importance as outcome of the PLS-DA, so one could wonder how significant the differences really are.

It would be also important to state if the high, medium and low exposed conditions follow certain dose-response characteristics or if the actual exposures to the different environmental toxins vary quite substantially between groups. The authors correlate heavy metal exposure to the selection of antibiotic resistance genes. Here the question is how well the metadata of subjects would exclude the possibility of confounding effects. It would also be helpful to perform experimental studies to validate these findings, since the selection of antibiotic resistance genes in response to heavy metal exposure seems an important hypothesis. In summary, the findings are really interesting but exclusively descriptive and the rationale to pick the presented differences is not clear.

Reviewer #1 (Remarks to the Author):

Reviewers Comments

This study investigates the impact of environmental pollution on the gut microbiome in a population from a highly polluted area in Campania, Italy. It presents novel findings showing that exposure to pollutants like dioxins and heavy metals alters gut microbiome composition and increases the abundance of microbial genes related to pollutant degradation and resistance. This adaptation suggests the gut microbiome's potential role in detoxifying harmful compounds, highlighting its significance in environmental health interactions and risk assessment models. The research underscores the urgent need for further exploration of the microbiome's capacity to counteract pollution's adverse health effects, potentially leading to the development of probiotic treatments.

Major Comments:

The differences in the microbiome betadiversity between the 3 populations is quite striking, I worry about other drivers beyond pollution – while I agree that the communities appear to be relative homogeneous. I would like to see a more robust analysis of the microbial differences using more appropriate statistical tests to account for structural differences between the cohorts (see below).

It would be good to determine if the microbiome is predictive of burden – and what those features are, i.e. Random Forest. Also, I would like to see some attempt to derive predictive assessment with AuROC analysis to determine if the microbial features are predictive of the pollutant burden. It would also be nice to see some attempt as using tools like mmvec (<https://www.nature.com/articles/s41592-019-0616-3>) to really determine if the pollutant levels in blood are associated with specific MAGs or pathways.

We thank the reviewer for these suggestions to improve our work. We performed mmvec to estimate correlations between the abundance of microbial taxa and metabolites (metals and dioxins). The results of the analysis are shown as Supplementary Figure S4, which highlights that two *Pseudomonas* mOTUs (phylum Pseudomonadota; ref_mOTU_v3_00154 and ref_mOTU_v3_00150), already identified as significantly more abundant in the MEDIUM group using Wilcoxon's tests (Supplementary Data S4), co-occur with heavy metal concentration (mercury -Hg - and antimony - Sb), while two mOTUs identified as *Ruminococcus gnavus* and unidentified *Lachnospiraceae* (ref_mOTU_v3_01594 and ext_mOTU_v3_16324) are negatively associated with dioxins (PCDD/Fs and DL-PCB) . We added these results in the main text (lines 160-166).

In addition, we carried out Random Forest classification based on microbiome taxonomic composition at species-level. The results are summarized in the table below. Indeed, we found that gut microbiome composition was able to discriminate subjects in the three groups with a good level of accuracy (>0.7 for all the comparisons). We added these details in the revised text (lines 143-159) and the most discriminant taxa were summarized as Supplementary Figure S3.

Class 1	Class 2	AUC
Low	Medium	0.77
Low	High	0.71
Medium	High	0.83

Minor Comments:

Ln 87-91 – this paragraph feels like it belongs earlier in the justification for this introduction.
 Ln 141 – this is the first report of ‘mOTUs’ so please define.

Done.

Ln 130-131 – “with clear clustering of the HIGH and LOW groups (Figure 1A)” – maybe I am missing something, but I do not see any ‘clear clustering in this ordination, if anything the communities looks fundamentally different to each other? Can you run this as Aitchinson’s distance and RobustPCA to see if these differences hold up for Fig. 1b?

Thanks for the suggestion. We calculated Robust Aitchinson’s distance, and the results were similar to those obtained using Jaccard’s. The violin plots summarizing these results are provided as Supplementary Figure S2 in the revised version of the manuscript.

Ln 133-142 – it is not clear in Fig 1 C and D what the ‘abundance metric’ is for the Y axis. And you really should not use Wilcoxon for compositional data. If you did a CLR-transform you could maybe argue that it works, but it’s better to look at log-ratios with something like ANCOM BC or SongBird? Also why do this at the Phylum level, why not run a proper OGU analysis (<https://journals.asm.org/doi/full/10.1128/msystems.00167-22>) and then use a log-ratio analysis to identify those taxa that are differentially proportionally different. Even better run Random Forest and find the taxa that best describe the differences between these communities. Also please note that all your data are proportional and so try and refrain from using the word ‘abundance’, unless you put relative in front of it.

Thanks for the suggestions. As reported above, we carried out Random Forest classification on microbiome taxonomic composition data and found a good discriminant power of the microbiome composition between the three groups. Results have been added in the text (lines 143-159) and most discriminant taxa have been summarized in supplementary figure S3. As reported in the text, several discriminant taxa from Random Forest classifier overlap with those reported in Suppl. Data S4.

Regarding phylum level analysis, we agree that it is not highly informative. However, we found interesting that even at a low level of taxonomic resolution, the differences among the three groups are so evident. Therefore, we decided to include phylum level differences in the main figures. In addition, we also provided differences at species level in the text (lines 138-156) and Supplementary Data S4. In the revised text, we added Random Forest classification on species-level taxonomic composition of the microbiome. These results are also reported in the revised text. We changed all the occurrences of “abundance” with “relative abundance” in the text and corrected Fig. 1C-D.

Reviewer #2 (Remarks to the Author):

In general, De Filippis and colleagues explored a fantastic cross-sectional population cohort in the Campania region of Italy covering different exposures to environmental pollution grouped in HIGH, MEDIUM and LOW. Fecal samples and blood samples were retrieved from 359 healthy subjects randomly selected from the SPES trial (4227 subjects). Trace elements and dioxins were determined in blood. Fecal microbiome profiling showed compositional changes in response to the level of pollution and identified bacterial genes associated with dioxin degradation and antibiotic resistance. In summary, the study addresses the important question to which extent the environment affects microbiome changes, and the presented study cohort is clearly designed and suitable to address this question. While the data give valuable insights, the study remains

descriptive and lacks validation.

Since only a sub-group of subjects in the SPES trial have been assigned to this study, additional validation seems possible in remaining smaller groups of subjects. Are compositional and functional changes reproducible and representative to the pollution categories. Would it be possible to validate response to selected chemicals in selected bacteria identified in the analysis?

We thank the reviewer for appreciating our work. The most suitable way to validate the results would be the assessment of the effect of selected chemicals in some of the bacteria identified as correlated with pollutants. This could be done in vitro or in pre-clinical trials, which would require the set up of new projects. Exploring in-vitro microbial dynamics exposing strains to chemicals would be interesting and would provide a validation of our results. However, this analysis would require a different study, as well as the isolation of specific strains from human gut. In addition, a new project (and additional fundings) would be necessary to sequence the fecal metagenomes of the remaining subjects enrolled in the SPES trial (about 3860) and validate the results on the whole cohort.

In Figure 1, the authors use PLS-DA analysis to show differences between high, medium and low exposed subjects. However, PLS-DA is suitable to identify features that best describe differences between groups rather than differences in microbiota composition per se. I wonder how the supervised and un-supervised PCoA analysis describes differences between these groups. Are the selected targets for further analysis justified by statistical selection or choices of the authors? The authors give no feature selection including feature importance as outcome of the PLS-DA, so one could wonder how significant the differences really are.

We thank the reviewer for the suggestion. The genera/species commented in the results are those significantly different among the three groups after Wilcoxon paired-tests (all the significant are reported as Supplementary Data S4). In addition, we added in the revised text machine-learning (Random Forest) classification based on microbiome taxonomic profiles and found a good predictive power of the microbiome composition. Data are reported in the revised text (lines 143-159) and the most discriminant taxa among the three groups are reported in Supplementary Figure S3.

Finally, according to the Reviewer's suggestion, we extracted the top 20 most important mOTUs from the PLS-DA, which are reported in the table below. Interestingly, all of them belong to the phylum Pseudomonadota, and those identified as *Pseudomonas* species showed a higher abundance in the MEDIUM group (most of them were already identified as statistically significant taxa by Wilcoxon's tests and reported in Supplementary Data S4). We added these results in the text (lines 140-145)

motu	Phylum	Species
ext_mOTU_v3_18094	Pseudomonadota	Phyllobacterium species incertae sedis
ref_mOTU_v3_00129	Pseudomonadota	Pseudomonas sp.
ref_mOTU_v3_00133	Pseudomonadota	Pseudomonas sp.
ref_mOTU_v3_00140	Pseudomonadota	Pseudomonas chlororaphis
ref_mOTU_v3_00146	Pseudomonadota	Pseudomonas sp.
ref_mOTU_v3_00149	Pseudomonadota	Pseudomonas sp.
ref_mOTU_v3_00150	Pseudomonadota	Pseudomonas sp.
ref_mOTU_v3_00152	Pseudomonadota	Pseudomonas moorei
ref_mOTU_v3_00153	Pseudomonadota	Pseudomonas sp.
ref_mOTU_v3_00154	Pseudomonadota	Pseudomonas fluorescens
ref_mOTU_v3_00161	Pseudomonadota	Pseudomonas umsongensis
ref_mOTU_v3_00162	Pseudomonadota	Pseudomonas silensiensis

ref_mOTU_v3_00163	Pseudomonadota	Pseudomonas fluorescens
ref_mOTU_v3_00164	Pseudomonadota	Pseudomonas sp.
ref_mOTU_v3_00167	Pseudomonadota	Pseudomonas sp.
ref_mOTU_v3_00168	Pseudomonadota	Pseudomonas koreensis
ref_mOTU_v3_00169	Pseudomonadota	Pseudomonas sp.
ref_mOTU_v3_00170	Pseudomonadota	Pseudomonas fluorescens
ref_mOTU_v3_00185	Pseudomonadota	Pseudomonas capeferrum
ref_mOTU_v3_01648	Pseudomonadota	Pantoea rodasii

It would be also important to state if the high, medium and low exposed conditions follow certain dose-response characteristics or if the actual exposures to the different environmental toxins vary quite substantially between groups. The authors correlate heavy metal exposure to the selection of antibiotic resistance genes. Here the question is how well the metadata of subjects would exclude the possibility of confounding effects. It would also be helpful to perform experimental studies to validate these findings, since the selection of antibiotic resistance genes in response to heavy metal exposure seems an important hypothesis. In summary, the findings are really interesting but exclusively descriptive and the rationale to pick the presented differences is not clear.

We thank the reviewer for this comment. The information about antibiotic use in the 6 months before recruitment was available for the whole SPES cohort and it was added in M&M and the metadata provided in Supplementary Table S1. As shown, only few subjects in each group declared the use of antibiotics and the occurrence of antibiotic use did not differ among the 3 groups as shown by Fisher's exact tests ($p > 0.05$).

In addition, the relationship between heavy metals exposure and acquisition of AMR genes was previously found in different environments (e.g., ref. 38-40), supporting these findings. Also in this case, ad-hoc designed experiments would be necessary to validate in-vitro or in animal models the observations and the dose-response effect.

REVIEWERS' COMMENTS

Reviewer #1 (Remarks to the Author):

Thank you for the edits. I am satisfied that the work is robust, rigorous and impactful.

Reviewer #2 (Remarks to the Author):

The authors sufficiently replied to my comments, specifically adding a bit more information to the compositional analysis. I agree that validation may be beyond the scope of this project.